# Colonization with *Escherichia coli* ST131-*H*30R (*H*30R) Corresponds with Increased Serum Anti-O25 IgG Levels and Decreased TNFα and IL-10 Responsiveness to *H*30R

**DOI:** 10.3390/pathogens12040603

**Published:** 2023-04-15

**Authors:** Brian D. Johnston, Connie Clabots, Tricia Bender, Stephen B. Porter, Germie van den Dobbelsteen, Jan Poolman, Paul Thuras, James R. Johnson

**Affiliations:** 1Departments of Medicine and Psychiatry, University of Minnesota, Minneapolis 55455, MN, USA; 2Infectious Diseases (111F), Minneapolis VA Health Care System, Minneapolis 55417, MN, USA; 3Janssen Vaccines & Prevention B.V., 2333 CN Leiden, The Netherlands

**Keywords:** *Escherichia coli* ST131-*H*30, intestinal colonization, IgG, TNFα, IL-10, antigen-stimulated cytokine release

## Abstract

An exceptional gut-colonizing ability may underlie the dramatic epidemiological success of the multidrug-resistant *H*30R subclone of *Escherichia coli* sequence type 131 (O25b:K+:H4). In order to inform the development of colonization-preventing measures, we studied systemic immune correlates of *H*30R intestinal colonization. Human volunteers’ fecal samples were screened for *H*30R by selective culture and PCR. Subjects were assessed by enzyme immunoassay for serum levels of anti-O25 IgG (representing *H*30R) and anti-O6 IgG (representing non-*H*30 *E. coli* generally), initially and for up to 14 months. Whole blood was tested for the antigen-stimulated release of IFNγ, TNFα, IL-4, IL-10, and IL-17 after incubation with *E. coli* strains JJ1886 (*H*30R; O25b:K+:H4) or CFT073 (non-*H*30; O6:K2:H1). Three main findings were obtained. First, *H*30R-colonized subjects had significantly higher anti-O25 IgG levels than controls, but similar anti-O6 IgG levels, suggesting an IgG response to *H*30R colonization. Second, anti-O25 and anti-O6 IgG levels were stable over time. Third, *H*30R-colonized subjects exhibited a lower TNFα and IL-10 release than controls in response to strain JJ1886 (*H*30R) relative to strain CFT073 (non-*H*30R), consistent with TNFα hypo-responsiveness to *H*30R possibly predisposing to *H*30R colonization. Thus, *H*30R-colonized hosts exhibit a sustained serum anti-O25 IgG response and an underlying deficit in TNFα responsiveness to *H*30R that could potentially be addressed for colonization prevention.

## 1. Introduction

The *Escherichia coli* ST131-*H*30R lineage (hereafter, *H*30R) emerged clinically beginning in approximately the year 2000 to become the main cause of multidrug-resistant *E. coli* infections worldwide and, in some studies, the leading *E. coli* lineage overall among human clinical isolates [1,2,3,4,5]. Its associated morbidity, mortality, and health care costs are enormous [6]. Hence, preventive measures against *H*30R are sorely needed.

For *H*30R, as for other gut-colonizing extraintestinal pathogens, intestinal colonization underlies extraintestinal infections [7,8,9]. Therefore, the prevention of *H*30R gut colonization could conceivably prevent both infection in the colonized host and dissemination of the strain to other hosts. As such, determinants of *H*30R gut colonization are important to elucidate.

Host immunity may factor importantly in *H*30R gut colonization. Gut bacteria can both elicit host immune responses and be acted on by them [10,11,12]. Therefore, immune elements, both humoral and cellular, could conceivably prevent and/or respond to *H*30R gut colonization.

Accordingly, in order to inform the possible development of immunity-based anti-colonization measures, we assessed immune response markers among human volunteers enrolled in a longitudinal, observational gut colonization surveillance study [13,14]. We focused on systemic immunity, both for logistical reasons and because it currently is more amenable to interventions than gut immunity. The main study goal was to use cross-sectional comparisons to test for differences in specific markers of humoral and cellular immunity to *H*30R between individuals who, at enrollment, either were or were not colonized intestinally with *H*30R. We assumed that if colonized individuals had comparatively reduced *H*30-specific immunity this would identify a possible point of intervention to prevent colonization, whereas if they had enhanced *H*30-specific immunity this would likely reflect a response to colonization. A second study goal was to use longitudinal comparisons to determine whether, among individuals persistently colonized with *H*30R, humoral immune responses to *H*30R change appreciably over time, especially in relation to a loss of colonization.

## 2. Materials and Methods

### 2.1. Subjects, Fecal Samples, and Strain Typing

As described elsewhere [13,14], subjects in the parent fecal surveillance study included U.S. military veterans under care at the Minneapolis Veterans Affairs Medical Center (MVAMC) and all of their available household members. Veterans were recruited prospectively from May 2014 through May 2018 by sending invitations for study participation to all newly discharged MVAMC inpatients and randomly selected outpatients. Veterans who agreed to participate were encouraged to refer all adult household members. All participating human subjects provided written informed consent. Fecal swabs were collected according to an Institutional-Review-Board-approved protocol. Swabs were mailed at room temperature in commercial transport medium to the research laboratory for processing, as described below.

Subjects whose initial fecal swab, when processed as described below, yielded *H*30R (with or without other *E. coli*) were offered serial fecal sampling. Serial sampling was performed monthly for six months, and then every three months until the subject declined further follow-up or the initial *H*30R strain was not detected in two consecutive samplings.

Fecal swabs (Ames, Becton Dickinson, Franklin Lakes, NJ, USA) were streaked to Gram-negative selective agar (Tergitol 7, Himedia Laboratory, Lincoln University, PA, USA), with and without ciprofloxacin (4 mg/L) (Supelco, Bellefonte, PA, USA), for overnight incubation at 37 °C [13,14]. Indole-positive (Kovac’s reagent: Millipore, Burlington, MA, USA), citrate-negative (citrate agar: Simmons, Remel, Lenexa, KS, USA) colonies with a characteristic *E. coli* morphology were regarded presumptively as *E. coli*. Up to 10 presumptive *E. coli* colonies per plate (as available) were screened for clonality by using random amplified polymorphic DNA (RAPD) analysis [15]. All isolates yielded an RAPD profile (not shown).

Using duplicate boiled lysates for template DNA and relevant positive and negative controls, one colony per unique RAPD profile per sample underwent PCR-based screening for ST131 [16]. ST131 isolates were screened by PCR for membership in the *H*30 subclone, which characteristically exhibits serotype O25b:K+:H4 [2,17,18,19]. Ciprofloxacin-resistant ST131-*H*30 strains were considered to represent the *H*30R subclone. (In previous work, such single-colony PCR-based screening of individual fecal isolates from ciprofloxacin-supplemented plates was > 90% sensitive for detecting the presence of *H*30R in the sample, in comparison with the most sensitive method available, which is PCR analysis of population DNA extracted from a broth culture or the inoculum zone of an agar plate culture (not shown)). For non-*H*30R *E. coli* isolates, phylogenetic group, ST, and O type were not assessed.

### 2.2. Blood Sample Collection

Adult fecal sample donors were invited to contribute a blood sample for immune marker testing. The goal was to recruit as many ST131-*H*30R-colonized subjects as possible and an approximately equal number of non-colonized controls. ST131-*H*30R-colonized subjects who consented to an initial blood draw were also invited to undergo up to four additional blood draws, timed to roughly correspond with their fecal sample donations. All blood donors provided one or more serum samples (serum tubes: Becton Dickinson), whereas a subset also provided a single whole-blood sample (heparin tubes: Becton Dickinson) for cytokine release analysis. (Although one study goal was to assess immune marker changes associated with loss of *H*30R colonization, all longitudinally followed *H*30R-colonized subjects remained colonized for the duration of follow-up, precluding such analyses.)

Sera for antibody testing were frozen at −80 °C, pending analysis. Whole blood was processed immediately for cytokine release, as described below, and the resulting plasma was stored at −80 °C, pending analysis.

### 2.3. O6 and O25b Serology

Serum levels of anti-O6 IgG and anti-O25 IgG were measured using an established enzyme immunoassay method [20,21]. Anti-O25 was used in relation to *H*30R (typically O25b-positive), whereas anti-O6 was used as a control, given the ubiquity of anti-O6 IgG in the human population [22] and the prevalence of O6 strains among human fecal *E. coli* [23]. For the enzyme immunoassay, the target antigens were O6 and O25 antigen-specific LPS, whereas the positive and negative controls were reference human sera with defined levels of anti-O6 and anti-O25 IgG (all provided by Janssen Vaccines & Prevention B.V., Leiden, The Netherlands).

For the immunoassays, 96-well microtiter plates (Corning Inc., Corning, NY, USA) were inoculated with O6 or O25 LPS (0.125 mg/mL in PBS (Sigma-Aldrich, Saint Louis, MO, USA) containing methylated bovine serum albumin (mBSA: Sigma-Aldrich, Saint Louis, MO, USA), incubated overnight at 4 °C, triple-washed with PBS containing 0.05% Tween-20 (PBS-T: Millipore, Burlington, MA, USA), blocked with 5% skimmed milk (Millipore) in PBS-T, and triple-washed with PBS-T. Serum samples (test and control) were diluted serially from 20-fold to 2,048,000-fold in PBS-T. Dilutions were added to separate wells of the LPS-sensitized trays, which were incubated for 1 h at room temperature (RT) and then triple-washed with PBS-T and inoculated with the secondary antibody, a mouse monoclonal anti-human IgG conjugated with horseradish peroxidase (Abcam, Waltham, MA, USA). Trays were triple-washed with PBS-T, inoculated with tetramethylbenzidine (Supelco), and incubated 15 min’ at RT. The reaction was stopped with 2 M sulfuric acid (Sigma-Aldrich) and absorbance was read at 450 nm (SpectraMax M5, Molecular Devices, San Jose, CA, USA). After subtraction of the average absorbance value for all blank wells, test sample absorbance values were used to determine the sample’s 50% effective concentration (EC50) using an online calculator (https://www.aatbio.com/tools/ec50-calculator) (URL accessed on 26 February 2020).

Sera from study subjects were tested in batches on three separate days, with internal controls included in each run. Duplicate testing, which, to conserve reagents, was limited to a randomly selected 14% subset of the test samples, documented high same-day reproducibility (not shown). Results for day 2 (27% of samples) differed substantially from results for days 1 and 3 and performed poorly in the EC50 calculator, and hence were excluded from statistical analysis.

### 2.4. Antigen Stimulation of Whole-Blood Samples

Antigen stimulation was performed using archetypal strains *E. coli* JJ1886 (ST131-*H*30R; O25b:K+:H4) and, as a control, CFT073 (ST73; O6:K2:H1). Notably, ST73, which is a leading ST among human commensal and clinical *E. coli* isolates, accounts for a large fraction of O6 *E. coli* [23,24]. Suspensions of freshly grown JJ1886 and CFT073 were UV-sterilized using a crosslinker (Stratalinker: Stratagene California, La Jolla, CA, USA) and were stored in 10,000 CFU-equivalent aliquots at 4 °C in PBS with 20% glycerol. Heparinized fresh whole blood from subjects was diluted 1:1 in PBS and dispensed into three 50 mL conical tubes (Nunc, Rochester, NY, USA) together with, per tube, a thawed aliquot of one of the challenge strains or an equal volume of PBS with 20% glycerol (Thermo-Scientific, Waltham, MA, USA) (negative control). Components were mixed gently. Tubes were left open and incubated for 18 h at 37 °C in a CO_2_ incubator (Napco, Brooklyn Park, MN, USA), then centrifuged gently to separate cells from plasma, which was aspirated immediately and stored at −80 °C. Immune cell subsets were not assessed.

### 2.5. Detection of Cytokines

Levels of IFNγ and TNFα (both: pro-inflammatory type 1 cytokines), IL-4 and IL-10 (both: anti-inflammatory type 2 cytokines), and IL-17A (a pro-inflammatory IL-17 family cytokine) were determined in plasma from whole blood that had been stimulated with bacteria–glycerol (i.e., JJ1886 or CFT073) or PBS–glycerol. Testing was performed at the University of Minnesota Cytokine Laboratory by using an automated multivalent, antibody-coated bead method (Milliplex MAP Kits and Luminex detection: EMD Millipore Corp, Billerica, MA, USA). Based on preliminary range-finding experiments, IFNγ, TNFα, and IL-10 were detected using (standard-sensitivity) HCYTOMAG-60K-03 trays, whereas IL-4 and IL-17A were detected using (high-sensitivity) HSTCMAG-28SK-02 trays. Undiluted test samples were run in parallel on the same day, in duplicate, assorted randomly, without operator knowledge of sample identity. Results for a given strain, in pg/mL, were the average of the duplicates.

### 2.6. Statistical Analysis

Anti-O6 and anti-O25 levels were analyzed using ln-transformed (EC50) values. Cytokine levels were analyzed for values from stimulation with JJ1886 (ST131-*H*30R; O25b:K+:H4) or CFT073 (ST73; O6:K2:H1) and for the ratio of those values, in each case after subtracting the sample’s PBS–glycerol control result. Because cytokine data were not distributed normally, nonparametric tests were used. Unpaired comparisons between colonized and control subjects were tested using two-tailed *t*-tests (for IgG levels) or the Mann–Whitney U test (for cytokine levels). Paired comparisons of anti-O6 and anti-O25 IgG levels within a given subject were tested using a matched pairs *t*-test. A repeated measures ANOVA was used to compare within-subject differences in anti-O6 and anti-O25 IgG levels between colonizers and controls. Correlation between different variables was tested using Spearman’s Rho. Assessment for temporal trends in antibody levels among colonized subjects who underwent serial sampling was tested using a mixed regression model testing for linear change over repeated observations within subject, fitting a model with a fixed effect for time (months) and a random intercept. All analyses were conducted using SPSS v. 27 (released 2020; IBM Corp., Armonk, NY, USA) using two-sided tests at alpha = 0.05.

## 3. Results

### 3.1. Characteristics of the Study Population

In both the serological substudy and the cytokine release substudy, subjects tended to be elderly (median age, 69 or 68 years), male (75% or 84%), and U.S. military veterans (73% or 82%) (Table 1). Within each substudy, the comparison groups (26 or 29 *H*30-colonized subjects; 26 or 32 non-colonized controls) did not differ significantly for any of these demographic variables.

### 3.2. Anti-O25 and Anti-O6 IgG Levels: Cross-Sectional Comparisons

According to cross-sectional comparisons involving a single initial serum sample per subject, anti-O6 IgG levels were comparable among *H*30R-colonized subjects (*n* = 26) and controls (*n* = 26) (Table 2, Figure 1A).

By contrast, anti-O25 IgG levels (Table 2, Figure 1A) and the anti-O25/anti-O6 IgG ratio (Table 2, Figure 1B) were significantly higher among *H*30R-colonized subjects than controls. Correspondingly, in paired comparisons of an individual subject’s anti-O25 IgG and anti-O6 IgG levels, among both colonized and control subjects, the anti-O25 level was significantly higher than the anti-O6 level (mean difference in O25 vs. O6 ln (EC50) values: colonized, 2.2 [*p* < 0.001]; controls, 0.68 [*p* = 0.03]). Notably, however, this difference was significantly greater among colonized subjects than controls (F (1,50) = 9.682, *p* = 0.003).

### 3.3. Anti-O25 and Anti-O6 IgG Levels: Longitudinal Comparisons

For serial serum IgG testing, 26 *H*30R-colonized subjects underwent longitudinal follow-up for up to 14 months; this yielded 66 total samples. The proportion of subjects that remained in follow-up declined progressively over time but was still 46% at nine months (Figure 2A). All subjects remained *H*30-colonized for the duration of follow-up.

Whereas anti-O6 and anti-O25 IgG levels varied greatly from sample to sample for a given subject (not shown), for the overall population, they were highly stable over time, with no evidence of a temporal trend (Figure 2B). Statistically, in fixed-effects mixed models, the estimated per month change for the anti-O25 ln (EC50) was 0.012 (95% CI: −0.015, 0.039) (*p* = 0.33); the estimated per month change for the anti-O6 ln (EC50) was 0.008 (95% CI: −0.008, 0.024) (*p* = 0.33); and the estimated per month change for the anti-O25/anti-O6 ln (EC50) ratio was −0.001 (95% CI: −0.008, 0.005) (*p* = 0.65). In each instance, the 95% CI was narrow and spanned zero.

### 3.4. Antigen-Triggered Cytokine Release: Cross-Sectional Comparisons

The antigen-triggered release of IFNγ, TNFα, IL-10, IL-4, and IL-17 was measured for 61 subjects (29 *H*30R-colonized, 32 controls) after the incubation of a single whole-blood sample with UV-killed, glycerol-stored cells of JJ1886 (ST131-*H*30R; O25b:K+:H4; representing *H*30) or, in parallel, CFT073 (ST73; O6:K2:H1; representing non-*H*30 *E. coli*). Incubation in parallel with PBS–glycerol was used to control for background cytokine release. After the subtraction of the PBS control value, levels of IL-4 and IL-17 were negligible (not shown); hence, statistical analysis was limited to IFNγ, TNFα, and IL-10.

According to within-subject paired comparisons, for each of the three assessable cytokines (IFNγ, TNFα, and IL-10), the extent of the triggered cytokine release was significantly greater after incubation with CFT073 than with JJ1886 (Table 3).

This relationship held both overall and separately among *H*30R-colonized and control subjects (for each comparison, *p* < 0.001, Wilcoxon signed ranks test) (Table 3).

As for comparisons between colonized and control subjects, IFNγ release showed no significant between-group differences or consistent trends, regardless of the stimulating strain (Table 3, Figure 3).

By contrast, TNFα and IL-10 release tended to be lower among colonized subjects compared with controls in response to JJ1886, but significantly higher (TNFα: *p* = 0.04) or comparable (IL-10) in response to CFT073. Accordingly, for TNFα and IL-10, the JJ1886/CFT073 ratio was statistically significantly lower—by approximately 35%—among *H*30R-colonized subjects than among controls, indicating that colonized subjects had a comparatively weaker TNFα and IL-10 responsiveness to JJ1886 than to CFT073 (Table 3, Figure 3).

### 3.5. Association of Immune Responses with Host Characteristics

None of the studied immune response markers—including O6 and O25 IgG levels, and IFNγ, TNFα, and IL-10 release after exposure to JJ1886 or CFT073—were significantly associated with any of the studied demographic variables (age, sex, veteran status). Likewise, no statistically significant correlations were detected between anti-O6 and anti-O25 IgG levels, between different cytokines, or between cytokine levels and anti-O6 or anti-O25 IgG levels (not shown).

## 4. Discussion

Here, in order to clarify the systemic immune correlates of intestinal colonization with *E. coli* ST131-*H*30R, we assessed selected humoral and cellular immune markers among *H*30R-colonized adults and, in parallel, concurrent non-colonized controls. We then made cross-sectional comparisons of colonized subjects and controls, and assessed colonized subjects longitudinally for changes in serum antibody levels over time. Our findings support three main conclusions. First, *H*30R colonization may stimulate a systemic anti-O25 IgG response, which, however, fails to eliminate colonization. Second, among *H*30R-colonized individuals, the anti-O25 IgG response is temporally stable over approximately one year. Third, a selectively reduced TNFα responsiveness to *H*30R strains relative to non-ST131 *E. coli* (as represented here by CFT073), which is accompanied by decreased IL-10 responses, may favor *H*30R colonization, suggesting that the boosting of such responsiveness might protect against colonization, although additional studies are needed to substantiate this.

The first main finding, i.e., the higher serum anti-O25 IgG levels among *H*30R-colonized subjects compared with controls despite these two group’s comparable anti-O6 IgG levels, may suggest a causal relationship involving the presence of (O25 antigen-expressing) *H*30R strains in the gut of colonized individuals. Although the direction of causation is unclear, it seems likely that colonization drives the antibody response. Since diverse gut-resident organisms can stimulate the production of cognate serum antibodies [25,26], a systemic anti-O25 IgG response to a gut-resident *H*30R strain would be unsurprising. This phenomenon may result from the subclinical translocation of colonizing strains, and/or relevant antigenic components thereof, from the gut lumen into the systemic compartment [25,26].

The fact that the *H*30R-colonized subjects were colonized with *H*30R despite having higher systemic anti-O25 IgG levels than controls would, on first consideration, suggest that such serum antibodies, although potentially effective in preventing invasive infection, do not prevent such colonization. Despite its intuitive appeal, however, this conclusion may be unwarranted. For example, it is conceivable that some—or even most—individuals who become colonized with *H*30R develop an IgG response that actually does clear the organism. Our study would have classified such individuals as controls by virtue of their having already cleared their *H*30R strain when assessed for colonization status. Our colonized group may thus represent a biased subset of ever-colonized individuals, one enriched with persons who either became colonized only recently, so have yet to develop a full colonization-clearing IgG response, or are inherently incapable of responding adequately to the colonizing strain. A future study could assess for incident *H*30R colonization in relation to serum IgG and other immune markers.

The second main finding, i.e., that mean anti-O25 and anti-O6 IgG levels were stable over time among *H*30R-colonized subjects, suggests that sustained colonization provides an ongoing antigenic stimulus for continued antibody production. The fact that all subjects in the longitudinal substudy remained *H*30-colonized throughout follow-up precluded the assessment of the immune correlates of a loss of colonization, which would require a different study design.

The observed exceptional persistence of *H*30R in the human gut, as also noted previously [14], has likely contributed to *H*30R’s epidemiologic success. The basis for such persistence is undefined, but may involve one or more of the ST131-associated accessory traits—classically regarded as virulence factors, but perhaps primarily colonization factors—that have been linked statistically with intestinal persistence [14]. These include genes associated with adherence (*iha, yfcV*), toxin production (*sat*), iron uptake (*iutA*, *fyuA*, *chuA*), capsules (group 2 capsules, K5 capsule), serum resistance (*traT*), and miscellaneous other functions (*usp*, *ompT*, *clbB*, *malX*). The possibility that interventions, including possibly vaccine-stimulated systemic or intestinal immunity, against such factors could decrease colonization with *H*30 warrants consideration.

The third main finding, i.e., that *H*30R-colonized subjects had a lower antigen-triggered release of TNFα and IL-10 from peripheral blood cells than controls in response to *H*30R strain JJ1886 relative to non-*H*30R strain CFT073 (ST73; O6:K2:H1), has potential translational implications; that is, although the basis for this phenomenon is unclear, we tentatively propose that an underlying reduced inflammatory reactivity specifically to *H*30R strains—as reflected in the colonized subjects’ blunted in vitro TNFα response to JJ1886—may increase the host receptivity to colonization by *H*30R strains. If this hypothesis is correct, then interventions that boost such responsiveness could conceivably be used to prevent or eliminate *H*30R colonization, thereby reducing the infection risk and interrupting transmission.

The observed blunted TNFα response may conceivably lead to a weaker counter-regulatory IL-10 response, potentially explaining these subjects’ comparatively lower IL-10 levels. To clarify the basis for this otherwise seemingly contradictory decreased expression of a pro-inflammatory and an anti-inflammatory cytokine, future work could characterize the phenotype of the cytokine-expressing immune cells.

A forward causal relationship between the host’s immune status and the gut microbiota has been called “inside-out” crosstalk, whereby host immunity shapes both the composition of the gut microbiota—even at the species level [10,11,27,28] —and its physiology [29]. This literature supports the notion that colonic inflammation, of whatever cause, leads to higher luminal concentrations of oxygen and nitrate, which inhibit the protective endogenous anaerobic population, thus favoring the outgrowth of facultative bacteria such as Enterobacteria, including *Salmonella* and pathogenic *E. coli* [10,11,27,28]; antimicrobial peptides may also play a regulatory role [10]. Since the present study did not assess the inflammatory state of the colonic milieu, it is uninformative regarding the relationship between gut inflammation and colonization with *H*30R.

The reverse-direction hypothesis, i.e., that gut colonization with *H*30R strains selectively down-regulates the host TNFα and IL-10 responsiveness to such strains, also warrants consideration. This phenomenon, if operative, would represent a form of so-called “outside-in” crosstalk, i.e., from the gut microbiota to the host immune system [10,11]. Notably, however, even if antecedent *H*30 colonization does down-regulate host TNFα and IL-10 responsiveness to *H*30, if such an acquired hypo-responsiveness facilitates sustained *H*30 colonization, in essence creating a positive feedback loop, then countermeasures that restore normal responsiveness could conceivably help to terminate colonization. This fits with the concept that the gut microbiota can modulate host cytokine pathways that are involved with gut homeostasis [12].

Clearly, the complex ecological phenomena under consideration here could also potentially involve other *E. coli* strains, non-*E. coli* gut microbiota components, host genetics, exogenous exposures such as diet and antibiotics, temporal shifts in any of these factors, or interactions among them [30,31,32]. For example, the presence and abundance of specific non-enterobacterial taxa in the gut microbiota correspond with the presence or loss of *H*30 colonization (unpublished data, JRJ). The present findings may inform future studies that address such added layers of complexity.

The study has limitations. First, the population size was relatively small. Nonetheless, the cross-sectional comparisons identified multiple statistically significant associations and the longitudinal assessments identified no trends that might become statistically significant with a larger sample. Second, subjects were mainly elderly male veterans, limiting generalizability. Third, the immunological assessment addressed a relatively narrow range of markers and only systemic acquired immunity. Conceivably, intestinal secretory IgA may protect against colonization with *H*30R, notwithstanding its doubtful effect on mucosal colonization with other extraintestinal pathogens [33,34]. Fourth, immune cell subsets were not assessed, and possible between-group differences in their distribution could conceivably have contributed to the observed between-group differences in cytokine production. Fifth, the study assessed only prevalent, not incident, colonization, and considered this as a binary variable, whereas its density or spatial distribution might be relevant. Sixth, the only non-ST131-*H*30R comparators were ST73 and O6 *E. coli*, and colonization with such strains was not assessed. Seventh, the study was observational and correlative. Eighth, the sustained colonization observed in the longitudinal cohort substudy precluded the assessment of the immune correlates of a loss of colonization.

In summary, this pioneering study of the systemic immune correlates of intestinal colonization with *E. coli* ST131-*H*30R suggests that such colonization is associated with both a sustained serum anti-O25 IgG response and reduced TNFα and IL-10 responsiveness to *H*30R, although the latter needs more work. These findings identify potential paths to pursue toward preventing colonization with the pandemic multidrug-resistant *H*30R lineage of *E. coli.*

## Figures and Tables

**Figure 1 pathogens-12-00603-f001:**
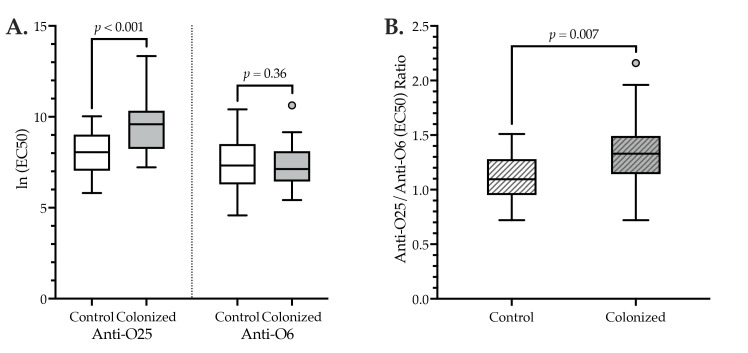
Cross-sectional anti-O25 and anti-O6 IgG levels (**A**) and their ratio (**B**) among *H*30R-colonized and control subjects. X-axes: controls (*n* = 26), *H*30R-colonized (*n* = 26). Boxes: bottom, 25th percentile; central black line, 50th percentile; top, 75th percentile. Whiskers: 1.5 x interquartile range. Small circle: outlier case. (**A**) Y-axis: individual-subject ln (EC50) values for anti-O25 IgG (left half of plot) and anti-O6 IgG (right half of plot). (**B**) Y-axis: ratio of (individual-subject) ln anti-O25 (EC50) IgG and ln anti-O6 (EC50) IgG values. *p* values by two-tailed *t*-test, assuming equal variance (confirmed by Levene’s test for equality of variances).

**Figure 2 pathogens-12-00603-f002:**
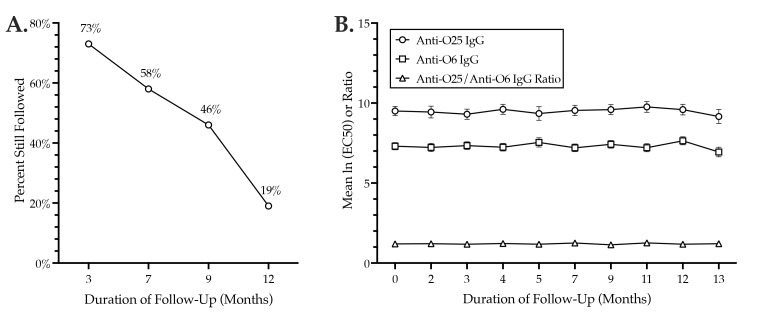
Anti-O25 and anti-O6 IgG levels during longitudinal follow-up of 26 *H*30R-colonized subjects. X-axes: duration of follow-up (months). (**A**) Percent of subjects over time with follow-up IgG data. Y-axis: percent of 26. (**B**) Serial serum IgG levels over 13 months. Y-axis: mean ln (EC50) values for anti-O25 IgG (circles) or anti-O6 IgG (squares) levels, or the ratio of these values (triangles). Error bars (anti-O25 and anti-O6 values only): standard error of the mean.

**Figure 3 pathogens-12-00603-f003:**
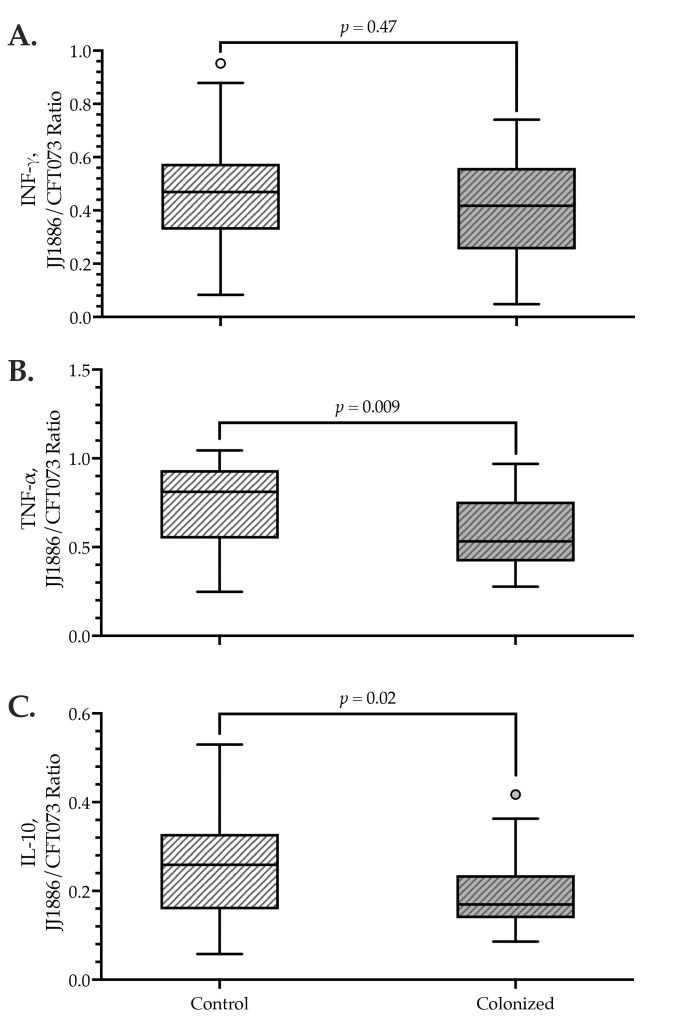
Ratio of antigen-stimulated cytokine levels after incubation with JJ1886 or CFT073 among *H*30R-colonized vs. control subjects. Plasma was separated and frozen for later cytokine analysis immediately after 18 h incubation of fresh whole blood with killed bacteria. (**A**) IFNγ. (**B**) IL-10, (**C**) TNFα. X-axes: controls (*n* = 32), *H*30R-colonized (*n* = 29). Y-axes: ratio of results after stimulation with JJ1886 (ST131-*H*30; O25b:K+:H4) or CT073 (ST73; O6:K2:H1). Small circles: outlier cases. *p* values by the Mann-Whitney U test.

**Table 1 pathogens-12-00603-t001:** Demographic characteristics of study subjects.

	Subject Characteristics within Subgroup ^a^					
	Serum IgG Substudy			Stimulated Cytokine Release Substudy		
Variable	Total (*n* = 52)	Control (*n* = 26)	*H*30R-Colonized (*n* = 26)	Total (*n* = 61)	Control (*n* = 32)	*H*30R-Colonized (*n* = 29)
Age, years: median (IQR)	69 (64,74)	73 (64,77)	68 (64,72)	68 (64,72)	69 (60.5,77)	68 (65,72)
Male, no. (column percent)	39 (75)	19 (73)	20 (77)	51 (84)	28 (88)	23 (79)
Veteran, no. (column percent)	38 (73)	18 (69)	20 (77)	50 (82)	27 (84)	23 (79)

^a^ For both substudies, all comparisons between colonized and control subjects yielded *p* > 0.05.

**Table 2 pathogens-12-00603-t002:** Cross-sectional serum anti-O6 and anti-O25b IgG levels among 26 *H*30R-colonized and 26 control subjects.

	ln (EC50 ^a^) or Ratio, Mean (SD ^b^)			
	Total	Control (*n* = 26)	Colonized (*n* = 26)	*p* Value ^c^, Colonized vs. Control
anti-O25b IgG, ln (EC50 ^a^)	8.8 (1.6)	8.0 (1.2)	9.5 (1.6)	<0.001
anti-O6 IgG, ln (EC50 ^a^)	7.3 (1.3)	7.3 (1.4)	7.3 (1.2)	0.36
anti-O25b/anti-O6 IgG ratio	1.2 (0.3)	1.1 (0.2)	1.3 (0.3)	0.007

^a^ EC50, 50% effective concentration. ^b^ SD, standard deviation. ^c^
*p* values by two-tailed *t*-test, assuming equal variance (confirmed by Levene’s test for equality of variances).

**Table 3 pathogens-12-00603-t003:** Cross-sectional whole-blood antigen-stimulated cytokine release among 29 *H*30R-colonized subjects and 32 control subjects.

	Cytokine Release after Stimulation with JJ1886 (ST131-*H*30R, O25b:K+:H4) ^a^			Cytokine Release after Stimulation with CFT073 (ST73, O6:K2:H1) ^a^			JJ1886/CFT073 Cytokine Release Ratio ^a^		
	Median (IQR ^b^)			Median (IQR ^b^)			Median (IQR ^b^)		
Variable	Control	*H*30R-Colonized	*p* Value ^c^	Control	*H*30R- colonized	*p* Value ^c^	Control	*H*30R-colonized	*p* Value ^c^
IFNγ (pg/mL) or JJ1886/CFT073 IFNγ ratio	184 (518) ^d^	194 (538) ^e^	0.61	426 (743) ^d^	442 (978) ^e^	0.58	0.50 (0.26)	0.44 (0.32)	0.47
TNFα (pg/mL) or JJ1886/CFT073 TNFα ratio	9353 (5429) ^f^	7804 (6204) ^g^	0.48	11,451 (2092) ^f^	12,546 (4606) ^g^	0.04	0.83 (0.39)	0.53 (0.34)	0.009
IL-10 (pg/mL) or JJ1886/CFT073 IL-10 ratio	623 (664)	408 (323)	0.06	2590 (2252)	2485 (1332)	0.87	0.26 (0.18)	0.17 (0.10)	0.02

^a^ Cytokine data (from plasma separated and frozen immediately after 18 h incubation of fresh whole blood with killed bacteria) are adjusted for the corresponding sample’s PBS blank. JJ1886/CFT073 cytokine release ratios were calculated for each sample individually. Group medians (as shown here) cannot be calculated based on these summary statistics. ^b^ IQR, interquartile range. IQR values greater than the corresponding median value indicate an extended upward skew of the data. ^c^
*p* values (from the Mann–Whitney U test) are for comparisons of colonized and control subjects. ^d,e,f,g^ For all comparisons of cytokine levels after stimulation with JJ1886 vs. CFT073, *p* < 0.001.

## Data Availability

The data presented in this study are available on request from the corresponding author. The data are not publicly available due to privacy and institutional ownership considerations.

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
