# Peer review of "Colonization with Escherichia coli ST131-H30R (H30R) Corresponds with Increased Serum Anti-O25 IgG Levels and Decreased TNFα and IL-10 Responsiveness to H30R"

_pathogens, 2023, doi:10.3390/pathogens12040603_

Round 1

Reviewer 1 Report

In this paper, the authors used cross-sectional comparisons to test for differences in specific markers of humoral and cellular immunity to H30R between individuals colonized intestinally with/without H30R. The authors demonstrated that H30R-colonized subjects had significantly higher anti-O25 IgG levels than controls, and found that H30R-colonized subjects exhibited less TNFα and IL-10 release than controls in response to strain JJ1886 (H30R). The data are clearly presented, experiments were appropriately controlled and the manuscript is clearly written.

Minor points :

In table 3 about IFNg (pg/mL) and JJ1886/CFT073 IFNg ratio

The number are 184 (518), 194 (538),   426 (743), 442 (978). Why the number in parentheses  is larger than the number without parentheses? How to calculate the ratio (0.50 and 0.44). They look calculated from the number in parentheses .

Author Response

Response to Reviewer 1

Reviewer comments (numbered) are shown sequentially below, each followed immediately by the corresponding author response, which is labeled "Response".

1. In table 3 about IFNg (pg/mL) and JJ1886/CFT073 IFNg ratio: The number are 184 (518), 194 (538),   426 (743), 442 (978). Why the number in parentheses is larger than the number without parentheses?

Response: The numbers in parentheses represent interquartile range (IQR) values. This is explained in the column headings and in footnote (b). To clarity why some of the numbers in parentheses (i.e., IQR values) are larger than the numbers without parentheses, we have extended footnote (b) by adding, "IQR values greater than the corresponding median value indicate an upward skew of the data."

2. In table 3 about IFNg (pg/mL) and JJ1886/CFT073 IFNg ratio: How to calculate the ratio (0.50 and 0.44). They look calculated from the number in parentheses .

Response: To clarify how these ratios were calculated, we have extended footnote (a) by adding, "JJ1886/CFT073 cytokine release ratios were calculated for each sample individually. Group medians (as shown here) cannot be calculated based on these summary statistics."

Reviewer 2 Report

The H30R E. coli lineage colonizes human hosts, which can lead to invasive infections. H30R is the most common cause of drug-resistant E. coli infections. The authors of this report assembled and evaluated a cohort of Minneapolis, MN veterans and their family members to investigate correlation between H30R colonization and host immune responses. The study presents evidence that H30R-colonized subjects had significantly higher anti-O25 IgG levels than controls, but similar anti-O6 IgG levels, observed stability of anti-O25 and anti-O6 IgG levels over time, and found that stimulation of whole blood from H30R-colonized resulted in lower TNFα and IL-10 production than was seen in controls. The findings are correlative rather than mechanistic and interpretations of the results are not reliable without further data. However, the data are reported in a robust manner and could prove useful for improving the design/execution of future studies.

Specific comments:

Fig 1A & 2B – Can the authors speculate on a reason for the higher anti-O25 titers (versus anti-O6) in control, non-colonized group?

Section 3.4, Table 3, and Figure 3 – The authors should specify in text and legends the timing of blood cell stimulation prior to collection of supernatants for cytokine assay.

Were immune cell proportions quantified in the blood samples prior to or after the LPS stimulation? Is there any evidence that the populations were similar or different in the control and H30R-colonized subjects? If there was no evaluation, this is a caveat the authors should mention when discussing results. Perhaps the weaker response to LPS in the H30R-colonized individuals reflects reduced abundance or phenotype of peripheral blood monocyte populations or subsets?

Discussion, lines 297-305 – An alternative explanation the authors may consider discussing is that serum IgG responses reflect responses elicited when E. coli cross the gut epithelium. Such Abs may protect only against invasive infection, vs intestinal colonization.

It would also be of interest for the authors to comment on why there was no observed colonization of control family members across the course of the study. Is this simply due to the study design being unable to assess this parameter? If so, in future efforts it may be of interest to modify the design to permit detection of such events and how the onset of colonization might affect immune parameters.

Author Response

Response to Reviewer 2

Reviewer comments (numbered) are shown sequentially below, each followed immediately by the corresponding author response, which is labeled "Response"

  1. Fig 1A & 2B – Can the authors speculate on a reason for the higher anti-O25 titers (versus anti-O6) in control, non-colonized group?

Response: The difference between O25 vs. O6 IgG levels among non-colonized subjects, as shown in Fig. 1A, was fairly small (mean, 0.68 ln[EC50] units) and of marginal statistical significance (P = 0.03), suggesting that it might not be real. If it was real, it could have been due to any of multiple possible phenomena, both biological and analytical. Given its uncertain validity, its multiple possible explanations, and the absence of relevant explanatory data, we are reluctant to speculate about it. By contrast, Fig. 2B addresses only colonized subjects, so is non-germane to control (non-colonized) subjects.

  1. Section 3.4, Table 3, and Figure 3 – The authors should specify in text and legends the timing of blood cell stimulation prior to collection of supernatants for cytokine assay.

Response: We now specify in the Methods text, a footnote under Table 3, and the Figure 3 legend that plasma was collected for cytokine analysis immediately after the end of the incubation of whole blood with killed bacterial cells.

  1. Were immune cell proportions quantified in the blood samples prior to or after the LPS stimulation? Is there any evidence that the populations were similar or different in the control and H30R-colonized subjects? If there was no evaluation, this is a caveat the authors should mention when discussing results. Perhaps the weaker response to LPS in the H30R-colonized individuals reflects reduced abundance or phenotype of peripheral blood monocyte populations or subsets?

Response: Immune cell proportions in blood samples were not assessed, so could not be compared between controls vs. colonized-subjects. We agree that between-group differences in the abundance of different immune cell subsets conceivably could have contributed to the observed between-group differences in cytokine levels. We have added this to Discussion, at the end of the first paragraph regarding the first main study finding ("This phenomenon may result from subclinical translocation of colonizing strains, and/or relevant antigenic components thereof, from the gut lumen into the systemic compartment [25,26]."), and list it as the fourth study limitation.

  1. Discussion, lines 297-305 – An alternative explanation the authors may consider discussing is that serum IgG responses reflect responses elicited when E. coli cross the gut epithelium. Such Abs may protect only against invasive infection, vs intestinal colonization.

Response: We agree, and have added this consideration to Discussion, in the second paragraph regarding the first main study finding ("...such serum antibodies, although potentially effective in preventing invasive infection, do not prevent such colonization.")

  1. It would also be of interest for the authors to comment on why there was no observed colonization of control family members across the course of the study. Is this simply due to the study design being unable to assess this parameter? If so, in future efforts it may be of interest to modify the design to permit detection of such events and how the onset of colonization might affect immune parameters.

Response: First, we agree that a study of the immune events surrounding acquisition of colonization with H30 would be of interest. We now address this explicitly in Discussion, at the end of the section on the first main finding ("A future study could assess for incident H30R colonization in relation to serum IgG and other immune markers"), and as part of the fifth listed study limitation (i.e., lack of attention to incident colonization).

Second, we are puzzled by the comment regarding the supposed absence of colonization of control family members across the course of the study. The study's longitudinal component involved only colonized subjects, not controls. As such, whether initially non-colonized (control) subjects became colonized across the course of study was not assessed. Hence, we cannot comment on this topic. 

Round 2

Reviewer 2 Report

Prior concerns have been addressed/clarified in the authors' response and revisions.